# Quantifying telomeric lncRNAs using PNA-labelled RNA-Flow FISH (RNA-Flow)

Iria González-Vasconcellos [1,2,3,6✉], María A. Cobos-Fernández[1,2,3,6], Michael J. Atkinson[4],
José Fernandez-Piqueras[1,2,3,5] & Javier Santos[1,2,3,5]

Here we present a method to detect and quantify long non-coding RNAs, in particular those related to telomeres. By coupling the specificity of a peptide nucleic acid (PNA) probe with flow cytometry we have quantified cellular levels of TERRA and TERC lncRNAs in culture cell lines and PBMCs. This easy-to-use method appointed RNA-Flow allows reliable lncRNA quantification with broad applications in basic research and clinical diagnostics. In addition, the staining protocol presented here was proven useful for the detection and quantification of such lncRNAs on unfixed cells using confocal microscopy.

[1] Genome Dynamics and Function Programme, Genome Decoding Research Unit, Centro de Biología Molecular Severo Ochoa (CBMSO), Madrid, Spain.
[2] Institute for Molecular Biology-IUBM (Autonomous University of Madrid), Madrid, Spain. [3] Biology Department, faculty of science. Autonomous University of Madrid (UAM). Canto Blanco Campus, Madrid, Spain. [4] Chair of Radiation Biology, Technical University of Munich, 81675 Munich, Germany. [5] Institute of Health Research Jiménez Díaz Foundation, Madrid 28040, Spain. [6] These authors contributed equally: Iria González-Vasconcellos, María A. Cobos-Fernández.
✉email: iria.gonzalez@cbm.csic.es

ncRNAs are defined as non-coding RNAs typically longer than 200 base pairs lacking protein-coding ability. LncRNAs have been postulated as biomarkers for the diagnosis and prognosis of various diseases such as cancer[1] and they have been indicated as new therapeutic targets[2,3]. Two of those candidates are the telomeric associated lncRNAs TERRA and TERC, whose levels are altered in cancer cells[4] and other diseases[5] and have been lately postulated as therapeutic targets in cancer[3,6].

The TERRA family of lncRNAs are initiated at a subtelomeric position with transcription progressing into the telomere. Hence, they are characterized by a unique short chromosome-specific 5′ sequence, followed by extended 5′-(UUAGGG)-3′ repeats. This lncRNA is characterized by 5′-(UUAGGG)-3′ repeats, only transcribed from the C-rich strand[7]. Comprehensive quantification of TERRA has remained elusive as the gold standard RT-qPCR can only target one individual TERRA transcript arising from one chromosomal end at a time. A simple and accurate assay for overall TERRA quantification was the motivation for the development of the technique presented here, appointed RNA-Flow. The RNA-Flow assay combines PNA probe labelling of RNA molecules with flow cytometric quantification based on previous DNA protocols[8–10] yet with specifics for RNA detection.

In this work, we demonstrate the efficacy of the RNA-Flow technique using PNA probes at detecting and quantifying TERRA and TERC lncRNAs both in cell lines and PBMCs extracted from human blood samples.

## Results and discussion

### RNA-Flow protocol singularities
The development of this method was possible because of the nature of TERRA LncRNA sequences being complementary to one of the telomeric strands and therefore also to the telomeric probes available. As such, the PNA TelC probe (AATCCC)n besides being complementary to telomeric DNA, is also complementary to the repetitive TERRA sequence as shown (Fig. 1a). Two key points are crucial to a successful analysis of RNA targets using this technique. Firstly, the use of an appropriate RNA inhibitor during the labelling. Two inhibitors were tested, Vanadyl Complex (New England Biolabs) and Protector RNase inhibitor (Roche). Whilst Vanadyl complexes proved unsuitable due to autofluorescence, the protector RNase inhibitor (Roche) was able to prevent RNA degradation without interfering with the quantification (Supplementary Fig. 1). Secondly, avoid DNA denaturation by maintaining a reaction temperature under 40 °C during the labelling process. At this temperature, the hybridization buffer with formamide does not contribute to DNA denaturation[11] favouring selective labelling of RNA strands. We tested RNA hybridisation temperatures ranging from 25 to 40 °C, to find the best mean fluorescence and percentage of stained cells at 40 °C (Supplementary Fig. 2). Also, PNA concentration and hybridisation times were tested for optimal RNA labelling. Overnight labelling with 300 nM TelC worked best for TERRA detection regarding the number of cells labelled and mean fluorescence intensity (Supplementary Fig. 3). Different RNase concentrations were tested to establish the right concentration to significant signal depletion (Supplementary Fig. 4). A cell number trial was performed using 0.25×10⁶; 0.5×10⁶; and 1×10⁶ cells with the established protocol for the RNA-Flow assay. No significant differences were found (Supplementary Fig. 5). Finally, an initial fixation step was tested using 2% formaldehyde for 10 min at room temperature prior to hybridisation in the buffer containing formamide. Results with our specific probes showed no significant differences when compared to those performed without the formaldehyde fixation. In fact, in the case of the probe used for TERRA detection, cells prefixed with formaldehyde showed a slight increase in nuclear background signal (Supplementary Fig. 6a and b). In the case of the TERC probe, no visible difference was observed between prefixed and non-prefixed cells with formaldehyde (Supplementary Fig. 7a). However, fixation can be of importance when gathering samples at different times, for specific lncRNAs or for specific probes. The fixation step was added as an optional step in the main protocol of the RNA-Flow assay and therefore its usage must be determined for each experimental requirement.

### Total TERRA quantification using RNA-Flow
The quantification of TERRA using RNA-Flow was performed in U2OS cells where the lncRNA was labelled with a Cy3-TelC PNA probe (named Cy3-TERRA). Representative cytometer peaks for Cy3-TERRA measurement in U2OS are depicted in Fig. 1b (red long dashed line) compared to unlabelled U2OS cells used as control (continuous black line). The specificity of the assay was tested by incubating the cells in the hybridisation buffer together with 100 µg of RNase. Results showed that the PNA signal was abolished by the RNase treatment and the cytometry fluorescence peak (grey short dashed line) looked similar to that of control unlabelled cells (Fig. 1b). The cellular integrity and TERRA staining were checked under confocal microscopy to find the nuclear-dotted pattern of the previously published TERRA staining performed by conventional RNA-FISH[12], but obtained this time on unfixed cells (Fig. 1c). The mean fluorescence intensity (MFI) of 3 technical replicates in U2OS were 125 ± 10.5 for unlabelled cells, 1235 ± 114 MFI for the Cy3-TERRA labelled cells and 96.6 ± 6.5 MFI for those Cy3-TERRA labelled cells treated with RNase (Fig. 1d). Moreover, the RNA-Flow analysis was repeated in two additional cell lines (MDAMB231 and HepG2), which interestingly showed cell-specific differences in TERRA levels (Fig. 1d). These results were validated using the gold standard RT-qPCR for the specific TERRA transcripts arising from chromosomes 10q and 15q (Fig. 1e). These data demonstrate the legitimacy of the technique when compared to the standard procedure but with the advantage of being able to perform total TERRA quantification. A telomere staining was performed in U2OS using DNA-flow with the same TelC probe for signal comparison and RNase reactivity. Telomeric DNA signal increased from the autofluorescence level of 110 ± 18.3 MFI in cells (black line) to a mean fluorescence intensity of 1254 ± 122.1 (red dashed line) in stained cells (Supplementary Fig. 8) but the telomeric DNA signal was not sensitive to the RNase treatment.

### RNA-Flow specificity demonstrated via TERRA knock-down with LNA GapmeRs
To further document the specificity of the assay at targeting this specific lncRNA, the TERRA transcripts in U2OS cells were knocked down by transient transfection with a specific antisense LNA GapmeR technology (Qiagen) as previously described[13]. TERRA knock down dynamics were assayed at 5, 24, and 48 h. The lncRNA expression reduction reached its lowest 24 h after transfection where TERRA levels were halved of those of the scramble control (Fig. 1f, g). This was largely recovered after 48 h (Fig. 1g). The expression levels were corroborated at the different time points by RT-qPCR of TERRA transcripts from 10q and 15q (Supplementary Fig. 9) and via confocal microscopy 24 h after transfection (Supplementary Fig. 10a, b). Both experiments corroborated the expression reduction seeing via flow cytometry.

### RNA-Flow detected TERRA expression differences throw-out the cell cycle
Knowing that TERRA levels vary throughout the cell cycle[14,15], we measured TERRA levels after hydroxyurea (HU) cell cycle synchronization of U2OS cells. Cells treated with

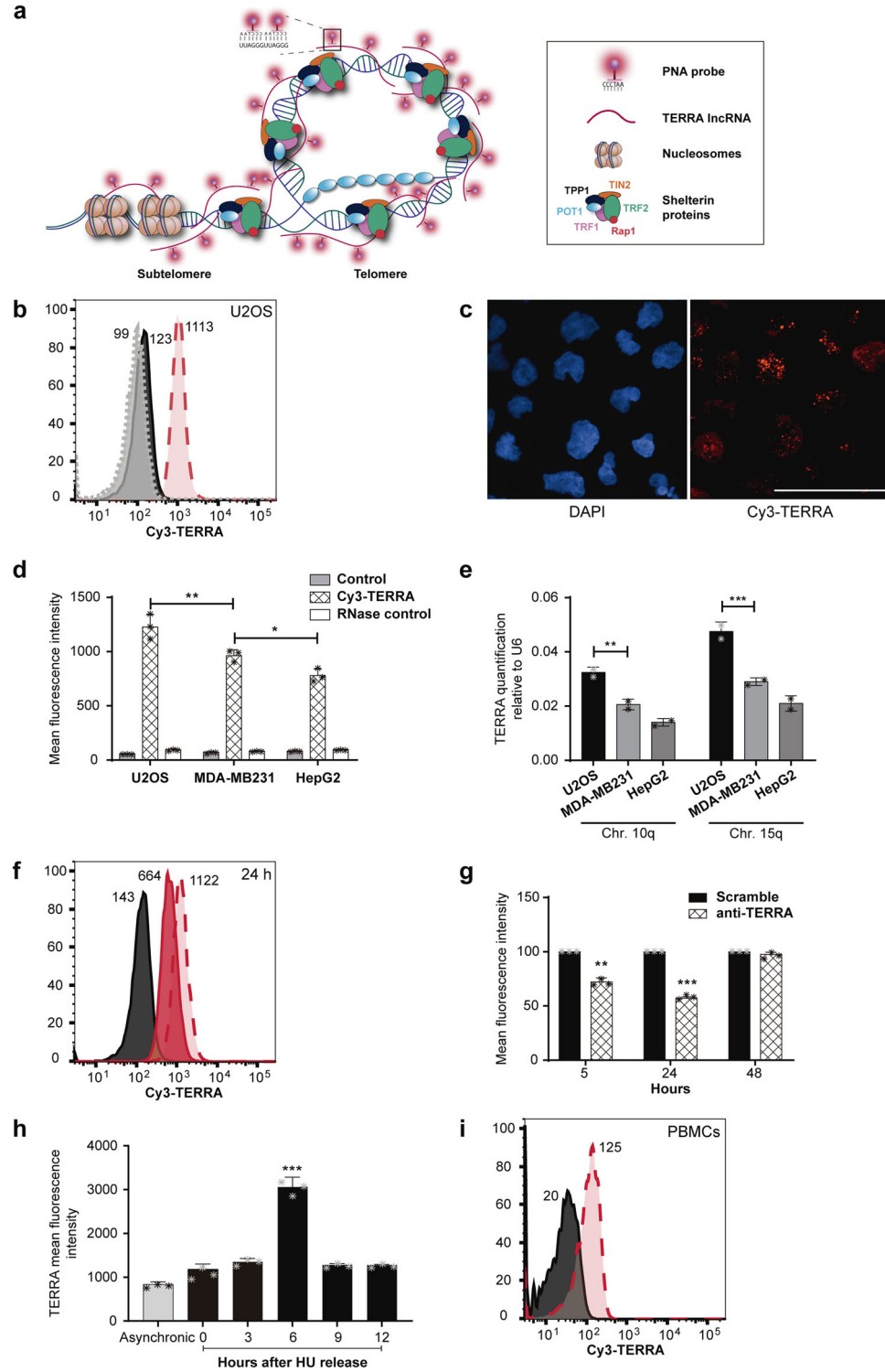

the drug accumulated in the late G1/ early S phase and samples were collected at 0, 6, 9, and 12 h after release as cells advanced through the cell cycle from G1 to late S phase (Supplementary Fig. 11). TERRA levels were assayed using RNA-Flow to determine variations during the late G1 and S phases. By using RNA-Flow we were able to determine that cells harvested 6 h after the release showed a significant increase in TERRA expression when compared to the other points assayed (Fig. 1h), demonstrating that the technique discriminates between cell cycle-specific expression differences of TERRA.

**TERRA analysis on PBMCs extracted from whole human samples using RNA-Flow.** To determine if this protocol could be applied to peripheral blood samples and therefore whether this protocol could be used for clinical proposes such as biomarker search, we analysed human peripheral blood mononuclear cells (PBMCs) from 3 healthy donors. In Fig. 1i, the results of a representative assay are shown. Quantification of 3 biological replicates showed a fluorescence intensity of $132 \pm 0.56$ MFI compared to a background of $20 \pm 0.89$ MFI. These significant results demonstrate the technique's effectiveness and potential in

**Fig. 1 TERRA quantification using RNA-Flow. a** Scheme of the TelC PNA probe hybridising TERRA lncRNA at telomeric chromatin. **b** Representative cytometry image of TERRA quantification (long dashed line filled in red peak) using RNA-Flow versus the unstained control (Black solid line peak) and the stained sample together with an RNase treatment (short dashed line filled with grey peak) in U2OS cells. **c** Confocal microscopy images of U2OS cells stained using the RNA-Flow technique. Left panel shows the DAPI staining demonstrating preserved nuclear morphology. Right panel, Cy3-TelC PNA probe detecting the dotted TERRA staining inside the nuclei. Displaying a region of interest (ROI) from Supplementary Fig. 6A after a Z-stack reconstruction (sum intensity projection). Scale bar 50 μm. **d** Quantification of 3 independent RNA-Flow experiments staining TERRA lncRNA in U2OS, MDA-MB231 and HepG2 cell lines. Grey bars show the unstained controls, checked bars show the mean of TERRA quantification and unfilled bars show the RNase control. Numbers represent the mean fluorescence intensity of 3 independent experiments. $**p \leq 0.01$ and $*p \leq 0.05$ using Student's $t$-test. **e** Validation of the RNA-Flow method by quantifying TERRA expression using RT-qPCR on human chromosomes 10q and 15q. Mean of 3 biological replicates using Student's $t$-test, $**p \leq 0.01$ and $***p \leq 0.001$. Pearson correlation indexes between the qPCR and the RNA-Flow: For chromosome 15q: $r = 0.9216$, $R^2 = 0.8494$ and a $p$-value of 0.0004 (***). For chromosome 10q: $r = 0.8695$, $R^2 = 0.7560$ and a $p$-value of 0.0023 (**). **f** Representative image of the RNA-Flow on TERRA quantification 24 h after transient knockdown using GapmeR technology. The black solid line peak represents the unstained control. The long-dashed line filled with light red peak represents the scramble control and the solid line filled in dark red represents the cells with TERRA knockdown. Numbers represent the mean fluorescence intensity. **g** TERRA quantification at different time points after GapmeR transfection in U2OS cells. Bars show the mean of 3 independent experiments. Solid bars are the scramble controls for each time point assayed. Checked bars are the TERRA quantification on knockdown cells. Mean of 3 biological replicates $**p \leq 0.01$ and $***p \leq 0.001$ using Student's $t$-test. **h** TERRA quantification on each cell cycle stage after HU block release. TERRA levels increase at a specific point of the S phase (6 h after release). Results show the mean of 3 independent experiments $***p \leq 0.001$ using the Student's $t$-test. **i** Representative image of RNA-Flow on PBMCs. The long-dashed line filled in red peak represents the mean fluorescence of TERRA versus their unstained control (black solid line peak). Numbers represent the mean fluorescence intensity.

clinical procedures. PBMCs isolation should be best performed within the first 24 h after blood extraction and analysed via RNA-flow either right after extraction or in fixed samples (optional fixation step with 2% formaldehyde).

**TERC quantification by RNA-Flow.** Further proof of the methodology was achieved by quantifying a second telomere-related lncRNA, TERC, one of the two main subunits of telomerase, highly related to telomeres and TERRA dynamics and of great importance in malignisation and inflammatory responses[16] as well as in the prognosis of human cancer and other diseases[6,17]. Currently, the analysis of TERC expression is conducted using RT-qPCR[17,18]. TERC expression was studied via RNA-Flow by using a newly developed PNA probe. A Cy3 labelled PNA probe (Supplementary Fig. 7b) was used for labelling TERC in MDA-MB231 cells. The representative image depicted in Fig. 2A shows the increase in fluorescence signal in the Cy3-TERC labelled cells (Red long dashed line) when compared to the unlabelled control cells (Black line). As shown for TERRA, the hybridization signal was abolished by including an RNase treatment (Grey short dashed line) (Fig. 2a). The cellular integrity and TERC staining were checked under confocal microscopy (Fig. 2b). The signal intensity of TERC was assayed in triplicates in three different cell lines, two telomerase positive cell lines (MDA-MB231 and HepG2) with different levels of TERC expression[16] and in human primary fibroblasts (hPF) with lower TERC levels. All the cell lines showed significantly different TERC levels assayed via RNA-Flow (Fig. 2c). The gold standard RT-qPCR ratified those differences (Supplementary Fig. 12). Pearson correlation indexes between the RNA-Flow and the gold standard qPCR for the TERC probe data: $r = 0.8389$, $R^2 = 0.7038$, and a $p$-value of 0.0047 (**)

**TERC detection and quantification on PBMCs extracted from whole human samples using RNA-Flow.** The analysis of this lncRNA on 3 samples of PBMCs via RNA-Flow demonstrated the detection of this RNA was possible in cells derived from blood samples (Fig. 2d) as previously shown via RT-qPCR[18,19]. Quantification in 3 biological samples in PBMCs showed positive cells with average fluorescence intensity of $146 \pm 3.2$ MFI when compared to control cells with an average mean of $45.7 \pm 2.4$ MFI.

**TERC knock-down with LNA GapmeRs detected by RNA-Flow.** To demonstrate the specific targeting of TERC by the PNA probe we developed an antisense LNA GapmeR (Qiagen) specifically designed against TERC lncRNA located towards the 5-prime end of TERC (Supplementary Fig. 7). Transiently transfecting MDA-MB231 cells with this GapmeR showed a reduction of the TERC lncRNA via RNA-Flow 24 h after transfection (Fig. 2e, f), and this expression reduction was confirmed by qPCR (Supplementary 13).

These studies demonstrate the legitimacy of the RNA-Flow technique for quantifying lncRNA expression and therefore this opens the flow cytometry advantages to RNA studies.

## Methods

**Cell culture.** All cell lines used in this study (U2OS, HEPG2, MDA-MB231, and hPF) were kept in culture in DMEM (Gibco) with 10% foetal bovine serum and 1% glutamine. Incubated at 37 °C in a humidified atmosphere with 5% $CO_2$. Cells were passaged when reaching confluence by using Trypsin/EDTA. hPF Human primary fibroblasts (hPF) were extracted from a piece of foetal muscle from an autopsy performed in a natural abortion. The participant provided written informed consent in accordance with the Declaration of Helsinki.

**PBMC extraction.** Whole blood preserved either in heparin or EDTA was used in this manuscript. Ficoll gradient was performed within the first 24 h after the extraction. Interphase with the PBMCs was collected. Cells were once washed in PBS and frozen in $N_2$. PBMCs cellular pellets were thawed and RNA-Flow was performed. Please note: Freshly extracted cells also work for the assay.

*Ethics.* PBMCs were obtained from healthy donors. Participants provided written consent in accordance with the Declaration of Helsinki. The ethical committee of the Autonomous University of Madrid approved the sample handling under the license CEI-98-1825.

**RNA-Flow.** $0.5 \times 10^6$ cells were collected via trypsinisation and washed in 500 μl of PBS. Please note that cell loads between $0.25 \times 10^6$ and $1 \times 10^6$ can be used without changing the outcome of the experiment. Cells were centrifuged at 260 g for 5 min at room temperature. The supernatant was discarded and cellular pellets were resuspended in 500 μl hybridisation buffer containing: 70% Formamide (Sigma-Aldrich), 20 mmol/l Tris buffer, pH 7, 0.5 μl (20 units) of the protector RNase inhibitor (Roche) and 300 nM of either Cy3-TelC for TERRA detection or Cy3-TERC PNA probe, Cy3-OO-AGCAGCTGACATTTTTTGTTTG (PANAGENE, South Korea). Please note: A fixation step can be performed before staining when required (see protocol below). Hybridisation took place overnight at 40 °C and gentle shaking (300 rpm in a thermoblock in the dark). Samples without PNA were used as a negative control. The RNase treatment was performed by adding into the labelling hybridisation buffer 50 μg crude mixture of RNases (DNase free) from bovine pancreas (Roche). Samples were centrifuged for 7 min at 1500 g. The supernatant was discarded and pellets were washed twice in Wash I: 70% formamide, with 0.1% tween-20 for 5 min with shaking (700 rpm) followed by

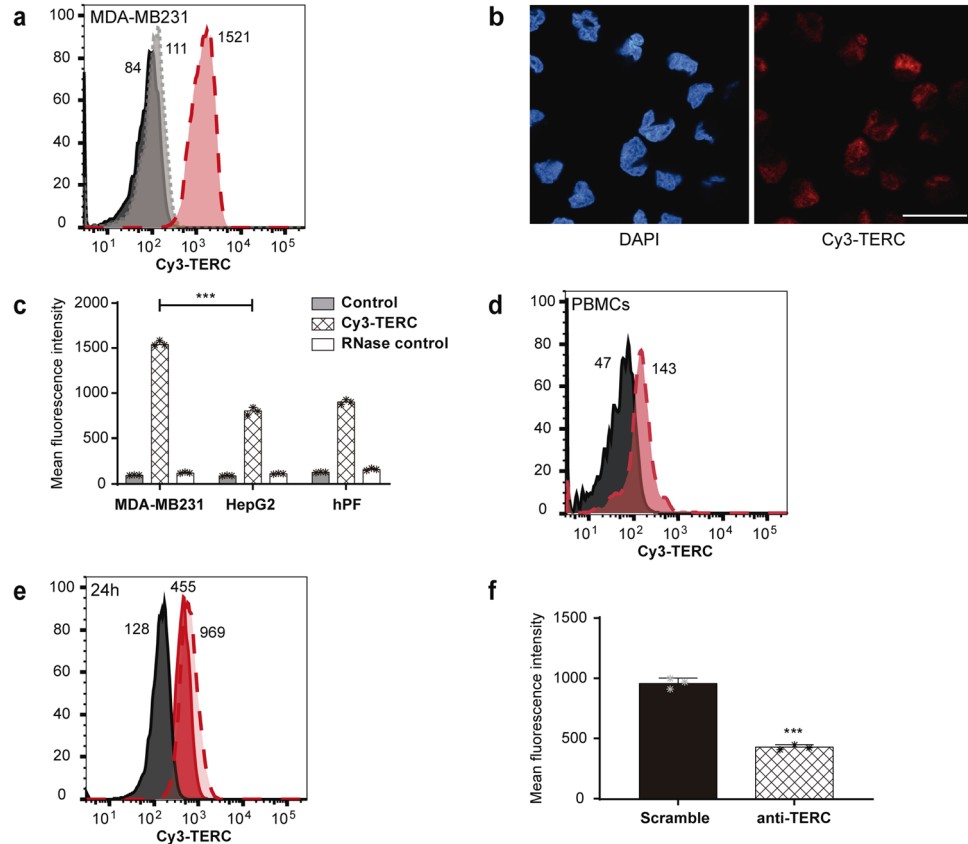

**Fig. 2 TERC quantification using RNA-Flow. a** Representative cytometry image of TERC quantification (long dashed line filled in red peak) using RNA-Flow versus the unstained control (Black solid line peak) and the stained sample together with an RNase treatment (short dashed line filled with grey peak) in MDA-MB231 cells. **b** Confocal microscopy images of MDA-MB231 cells stained using the RNA-Flow technique. Left panel shows the DAPI staining demonstrating the preserved nuclear morphology. Right panel, Cy3-TERC PNA probe detecting TERC lncRNA located all over the cell but most intense inside the nuclei. Displaying a Z-stack reconstruction (sum intensity projection). Scale bar is 25 μm. **c** Quantification of 3 independent RNA-Flow experiments staining TERRA lncRNA on MDA-MB231, HepG2 cell lines, and human primary fibroblasts (hPF). Grey bars show the unstained controls, checked bars show the mean of TERC quantification and unfilled bars show the control with RNase. Numbers represent the mean fluorescence intensity. ***$p \leq 0.0001$ using Student's *t*-test. **d** Representative image of RNA-Flow on PBMCs. Long dashed line filled in red peak represent the mean fluorescence of the TERC versus the unstained control (black solid line peak). Numbers represent the mean fluorescence intensity. **e** Representative image of the RNA-Flow on TERC quantification 24 h after transient knockdown of the lncRNA using GapmeR technology. Black solid line peak represents the unstained control. Long dashed line filled with light red peak represents the scramble control and the solid line filled with red represents the TERC levels after knockdown. Numbers represent the mean fluorescence intensity. **f** TERC quantification 24 h after GapmeR transfection in MDA-MB231 cells. The solid bar is the scramble control. The checked bar is the TERC quantification after knockdown. Mean of 3 biological replicates ***$p \leq 0.001$ using Student's *t*-test.

centrifugations for 7 min at 1500 *g*. The supernatant was discarded and pellets were rinsed in wash buffer II containing PBS with 0.1% tween-20 for 5 min at room temperature. Centrifugation was performed for 7 min at 900 *g*. The supernatant was removed and pellets were resuspended in 500 μl of PBS and transferred into cytometer tubes. Results were obtained in a FACSCanto A Flow Cytometer (Becton Dickinson Biosciences) gated for the different populations (Supplementary Fig. 14) and analysed with the FlowJo software (FlowJo, Ashland, OR).

*Optional fixation step when required.* Fixation with 2% formaldehyde can be performed prior to hybridisation. Here is the protocol we found best suited. Cell pellets were resuspended in 2% formaldehyde in 1 x PBS for 10 min at room temperature. Continue by adding 2 volumes of 1 x PBS to dilute the formaldehyde and centrifuge at 260 *g* for 5 min. Discard the supernatant and wash with 1 ml of PBS and centrifuge at 260 *g* for 5 min. Proceed by adding the hybridisation buffer as described above.

**Confocal imaging of RNA-Flow samples**. 0.5 x 10⁶ cells were collected via trypsinisation and washed in 500 μl of PBS. Cells were centrifuged at 260 *g* for 5 min at room temperature. The supernatant was discarded and cellular pellets were resuspended in 500 μl hybridisation buffer containing: 70% Formamide (Sigma-Aldrich), 20 mmol/l Tris buffer, pH 7, 0.5 μl (20 units) of the protector RNase inhibitor (Roche) and 300 nM of either Cy3-TelC for TERRA detection or Cy3-TERC PNA probe, Cy3-OO-AGCAGCTGACATTTTTTGTTTG (PANAGENE, South Korea). Hybridisation took place overnight at 40 °C and gentle shaking (300

rpm in a thermoblock in the dark). After staining, all cells were washed twice in wash buffer I containing 70% formamide, with 0.1% tween-20 followed by one extra wash in wash buffer II containing PBS with 0.1% tween-20. Cells were stained with DAPI in a 1:5000 PBS solution for 5 min at room temperature in the dark, followed by a PBS wash and centrifugation for 5 min at 260 *g*. The cell pellet was resuspended in a final volume of 150 μl of PBS and transferred into a glass chamber Glass bottom microwell dish 35 mm petri dish and 20 mm microwell (MatTek life sciences), followed by 15 min at room temperature to settle down and after chambers were placed under the microscope. Z-stacks of 80 images were captured in 0.47 μm sections across the entire nucleus with a Nikon A1R laser scanning confocal microscope (Nikon) using a 60X/1.4 Plan-Apochromat objective. Acquired Z-stacks were reconstructed (sum intensity projection) and analysed using Fiji v1.53.

**Telomere length measurement by flow cytometry (DNA-Flow)**. 0.5 x 10⁶ cells were resuspended in 500 μL hybridisation buffer containing: 70% formamide (Sigma-Aldrich) and 20 mmol/L Tris buffer, pH 7. Denaturation was performed at 80 degrees for 10 min followed by the addition of 300 nM PNA probe and 50 μg of RNase (Roche) and an overnight incubation at 40 degrees with gentle shaking (300 rpm). After staining, all cells were washed twice in wash buffer I containing 70% formamide, with 0.1% tween-20 followed by one extra wash in wash buffer II containing PBS with 0.1% tween-20. Cells were resuspended in 500 μL PBS for analysis. Results were obtained in a FACSCanto A Flow Cytometer (Becton Dickinson Biosciences) and analysed with the FlowJo (FlowJo, Ashland, OR) software.

**Propidium iodide labelling for FACS analysis of the cell cycle**. Cells were trypsinised and 1.5 x 10^6 cells were washed with 1 ml PBS and spun down at 400 $g$ for 5 min. 1 ml of 70% ethanol (cold) was added drop by drop to the cellular pellet while gently vortexing and left overnight at −20 °C. The following day fixed cells were centrifuged at 1500 $g$ for 5 min. Cellular pellet was washed twice in 1 ml PBS, centrifuged at 400 $g$ for 5 min. Clean pellet was then resuspended in 0.5 ml of PI/RNase staining buffer (BD Pharmingen). Incubate for 30 min at room temperature in the dark. Results were obtained in a FACSCanto A Flow Cytometer (Becton Dickinson Biosciences) and analysed with the FlowJo (FlowJo, Ashland, OR) software.

**TERRA RT-qPCR**. Total RNA was extracted with the High Pure RNA Tissue Kit (Roche diagnostics). RNA was treated with RNase-Free DNase (Qiagen) for 1 h, cleaned, and concentrated with RNA Clean & Concentrator™ (Zymo Research) with an additional in-column DNase treatment made before elution. Specific reverse transcription of TERRA and the U6 control RNA were performed using 10 μM of the TERRA specific oligonucleotide (CCCTAA)$_5$ or 1 μM of the U6 oligonucleotide GAACTCGAGTTTGCGTGTCATCCTTGCGC respectively. A qPCR was performed with 3 μl of this newly synthesized cDNA using GoTaq qPCR Master Mix (Promega), in a final volume of 20 μl. The following primers were used for the amplification of the human subtelomeric region from chromosome 10q (forward: GAATCCTGCGCACCGAGAT, reverse CTGCAC TTG AACCCTGCAATAC). For TERC the primers (forward: TCTAACCCTAACTG AGAAGGGCGTAG, reverse GTTTGCTCTAGAATGAACGGTGGAAG) were used, and U6 as a housekeeping gene.

**GapmeR transfection for TERRA/TERC knockdown**. Transient in vitro knockdown of TERRA and TERC in U2OS and MDA-MB-231 respectively was achieved using 15 nM modified antisense oligonucleotide GapmeRs, (Qiagen, formerly Exiqon) either in U2OS (TERRA) or MDA-MB231 (TERC). 300,000 cells were seeded in 6 well plates in 2 ml of complete medium. The following day, media was replaced by fresh media before transfection. Cells were transfected with lipofectamine 3000 according to the manufacturer's protocol. In brief, in one tube 7.5 μl of lipofectamine 3000 were diluted in 125 μl of Opti-MEM medium. In another tube 150 pmol of GapmeR (either scramble, anti-TERRA or anti-TERC) were diluted in 125 μl of Opti-MEM medium. Both tubes were mixed 1:1 and incubated 15 min at room temperature. This 250 μl of DNA-lipid complex are added drop by drop to the cell culture. Cells were incubated at 37 °C in a humidified atmosphere with 5% $CO_2$ and collected at the desired time points. Scramble LNA GapmeR scramble :5′CACGTCTAT ACACCAC 3′; LNA GapmeR anti-TERRA: 5′TAACCCTAACCCTAA 3′ and LNA GapmeR anti-TERC: 5′CTCTAGAATGAACGGT 3′(Quiagen).

**HU cell cycle block**. 250,000 cells were seeded in a 6 well plate in 2 ml complete medium. Hydroxyurea stock solution (500 mM) was prepared fresh before addition to the culture and filtered sterilised with a 0.2 μm pore size filter. Mix 50 ml of complete medium with 400 μL of filter-sterilized HU stock solution for a final HU concentration of 4 mM. Remove medium from all wells except those needed for defining FACS settings and replace with a freshly prepared 4 mM HU-containing complete medium (2 ml/well). Incubate cells for 24 h in a HU-containing medium at 37 °C in a humidified atmosphere with 5% $CO_2$. Remove HU-containing medium from wells and rinse wells twice with pre-warmed 1x PBS (2 ml each time). Add 2 ml of complete medium per well. Collect asynchronous sample and 0 h time point (synchronous). Place remaining wells in the incubator and collect at the desired time points (3, 6, 9, and 12 h).

**Statistics and reproducibility**. Statistical analyses (Student´s $t$-tests and Pearson correlation analyses) were performed using the GraphPad software (GraphPad Software Inc, California, USA). Analysis was performed in triplicates and the significance was represented using the $p$-value as follows: (*) $p \leq 0.05$; (**) $p \leq 0.01$; (***) $p \leq 0.001$; (****) $p \leq 0.0001$. Analysis was performed in triplicates.

**Reporting summary**. Further information on research design is available in the Nature Research Reporting Summary linked to this article.

## Data availability

All data generated or analysed during this study are included in this published article (and in Supplementary Data 1).

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

## Acknowledgements

The authors would like to thank the Cytometry and Cell Culture services at the Center of Molecular Biology "Severo Ochoa" of Madrid, (CBMSO). Isabel Sastre for the technical support and for sharing the PBMCs. Dr. Laura González Sanchez for her help in designing the figures. We also thank the microscopy service at the CBMSO, especially Carlos Gallego García y Francisco J. Vega Sabugo for their valuable input on confocal image acquisition and analysis. This work was financed by grants from the Spanish Ministry of Science, Innovation, and Universities (MCIU), (RTI2018- 093330-B-I00; MCIU/FEDER, EU), Ramón Areces Foundation (CIVP19S7917); Autonomous Community of Madrid, Spain (B2017/BMD-3778; LINFOMAS-CM); and the Spanish Association Against Cancer (AECC, 2018; PROYE18054PIRI). The Spanish Ministry and the Juan de la Cierva fellowship program must be acknowledged for the contract of I.G.V. (IJCI-2016-29155). Institutional grants from the Ramón Areces Foundation and Bank of Santander to the CBMSO are also acknowledged.

## Author contributions

I.G.V. and M.A.C.F. conceived the idea and carried out the experiments I.G.V.; M.A.C.F.; J.F.P.; and J.S.H. performed data analysis and evaluation for the first use. I.G.V. wrote the manuscript; M.J.A. and J.F.P. reviewed the manuscript.

## Competing interests

The authors declare the following competing interests: The authors have submitted a provisional patent application that is based on the method described in the manuscript (Ref: 906340).

## Additional information

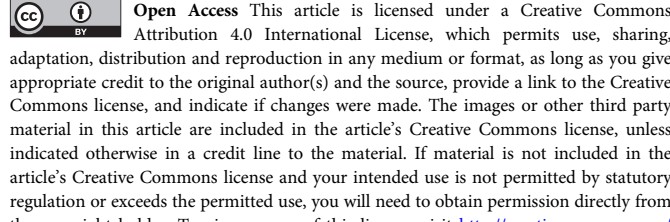

