## [Peer Review File · Communications Biology]

Reviewers' comments:

Reviewer #1 (Remarks to the Author):

In this manuscript, the authors described a novel method termed RNA flow to quantitate the amount of TERRA lncRNA, which is transcribed from (sub) telomeric regions of chromosomes. While the detection methods is of some interest, the authors fail to demonstrate specificity of the signals detected by the probes. I am not convinced if the signals come from the hybridized probes or trapped by non-specific RNA-PNA interactions.

1. Conventional hybridization-based in situ detection of RNA rely on highly stringent hybridization and washing conditions, which ALWAYS require fixation of the cells, as far as I know. The authors directly add probes diluted in a buffer containing a high concentration of formaldehyde without any salt. This is somewhat strange because they actually perform in situ hybridization using literally "dead" cells without fixation. More careful description and control experiments should be required. At least they should show high-magnification images of the labelled cells to confirm telomere-specific signals.

2. RNase treatment is not enough to confirm specificity of the probes. They should show decrease of telomere signals upon knock-down of TERRA using conventional high-resolution microscopic images.

3. The manuscript does not follow standard scientific format, lacking clear introduction and result sections. I understand Comm Biol accept wide variety of format at a initial submission, but the current format is not acceptable, at least for me.

Reviewer #2 (Remarks to the Author):

González-Vasconcellos et al. described a new assay for quantifying intracellular levels of d lncRNAs TERRA and TERC using flow cytometry and peptide nucleic acid probes. They provided solid demonstration of the assay's functionality. I only have two questions/suggestions for minor revision. Once these questions are addressed (if possible), the article will make an interesting read for the field of scientists working on lncRNA biology.

1. The authors used 0.5×10^6 cells for experiment. Could you comment on the effect of using higher or lower cell load?

2. The assay's applicability to peripheral blood cells can be of clinical utility. Could you describe in details how to process the blood sample: What anticoagulatives are compatible with the assay? What's the maximal acceptable transit time from blood draw to assay initiation (i.e. can leftover blood from say routine CBC or biochemistry tests be used)? How you retrieved the cells from a tube of blood?

Some typos were noted but did not affect the overall interpretation of the results:

1. Page 2, line 71: "RNase inhibitor" instead of "RNA inhibitor".

2. Page 16, line 573: "RNase concentrations" instead of "RNA concentrations"

3. Page 16, line 556: the sentence is grammatically confusing: "And making sure the reduction of the melting point due to the formamide in the hybridisation buffer does not infer cross reactions with the DNA."

Reviewer #3 (Remarks to the Author):

The study by Gonzalez-Vasconcellos et al. presents a new RNA-flow method using PNA probes to detect and quantify TERRA and TERC RNA by flow cytometry. They optimized the hybridization conditions, RNase inhibitors, and probe concentration. Three cell lines (U2OS, MDAMB231, and HepG2) were initially tested using this method and the results were consistent with the qRT-PCR data. They nicely showed that the reduction of TERRA RNA by knockdown can be confirmed using

this method. TERRA expression during cell cycle progression was analyzed by RNA-flow and it displayed an increase of TERRA level in the mid-S phase. They demonstrated that this method can apply to human periphery blood mononuclear cells (PBMC). Similar experiments were also performed for TERC RNA quantification using a Tel-G PNA probe. This RNA-flow method could be very beneficial for clinical assay and basic research. However, a high probe concentration used in the hybridization and no pause points in the protocol can be a drawback. The major concerns below are needed to be addressed before publication.

Major concerns

1. Does this RNA flow method have to be done in freshly collected cells? Any steps can be paused before hybridization?
2. The optimal probe concentration is 300 nM, which is quite high for RNA flow. The authors collected cells and performed hybridization without fixation and permeabilization steps, thereby the probes may not efficiently enter the cell. Duckworth et al., (Nature protocol, 2019) have described a method to detect RNA and proteins using flow cytometry and the probe concentration is in the range from pM to nM. Porichis et al., (Nature Comm, 2014) also used a branched DNA method to amplify RNA signals to detect RNA by flow cytometry (now is available in ThermoFisher Scientific, called PrimeFlow RNA assay). Both methods include fixation and permeabilization steps. Do the authors try the fixation and permeabilization steps before hybridization? Can authors compare their method and others' RNA flow methods to show its advantages and disadvantages?
3. PNA probe (TelG) for TERC can potentially target antisense TERRA transcripts. Why do the authors use specific PNA probes for TERC? The TelG probe only contains 11 nucleotides complementary to TERC. Is this specific enough for TERC RNA? Or this should be discussed.

Minor concerns

1. "RNase" is commonly used instead of RNAse. Please replace it throughout the manuscript. Replace "DNase" with "DNase" throughout the manuscript.
2. Please write in a symbol for the Celsius scale to replace "degrees" throughout the whole manuscript
3. Which RNase was used in Supplementary 4, 5? Please specify it. RNase A or a cocktail of RNases?
4. Line 280" there is a typo in 250.000 cells. It should be 250,000 cells.
5. Line 113: Missed a comma in the sentence. To further document the specificity at targeting this specific lncRNA, the TERRA transcripts in U2OS cells were
6. Line 187: a typo in "fetal" bovine serum

Reviewers comments:

Reviewer #1 (Remarks to the Author):

In this manuscript, the authors described a novel method termed RNA flow to quantitate the amount of TERRA lncRNA, which is transcribed from (sub) telomeric regions of chromosomes. While the detection methods is of some interest, the authors fail to demonstrate specificity of the signals detected by the probes. I am not convinced if the signals come from the hybridized probes or trapped by non-specific RNA-PNA interactions.

We thank the reviewer for the useful comments addressed in this letter. We hope our review, extra experiments and explanations clarify his/her doubts. We proceed to address her/his concerns one by one.

Conventional hybridization-based in situ detection of RNA rely on highly stringent hybridization and washing conditions, which ALWAYS require fixation of the cells, as far as I know. The authors directly add probes diluted in a buffer containing a high concentration of formaldehyde without any salt. This is somewhat strange because they actually perform in situ hybridization using literally "dead" cells without fixation. More careful description and control experiments should be required. At least they should show high-magnification images of the labelled cells to confirm telomere-specific signals.

Firstly, we would like to clarify that we use 70% formamide (not formaldehyde). Formamide, according to the American Chemical Society, is used to stabilize RNA and often used in tissue preservation. In fact, the telomeric Flow-FISH in DNA is also performed with just formamide at 70%. Together with the formamide in our protocol, we used the RNase inhibitor to help preserve RNA structures which we found key to obtaining reproducible results.

We however checked the cells right after staining (following the RNA-Flow protocol without any extra fixation, just the cell suspension we used for cytometry) in the confocal microscope and we show the picture below. The DAPI staining shows intact nuclear structure without DNA fragmentation or nuclei bursting and the TERRA signals are perfectly visible within the nuclei demonstrating correct cellular permeabilization and targeting of TERRA. In fact, these results are very similar to those obtained using conventional RNA-FISH.

Figure 1: U2OS cell stained with the RNA-Flow protocol seen under the confocal microscope. The DAPI image showed intact nuclei. PNA staining of TERRA transcripts with Cy3 show a dotted pattern within the nuclei with different sizes within each nuclei. The image is a representative Z-stag (stag 25 in the middle of the nuclei).

However, to eliminate any shadow of doubt about the protocol, we performed the RNA-Flow experiment after fixation with 2% and 4% PFA for 10 minutes prior to the PNA staining. We found a slight improvement on the mean fluorescence after 2% Formaldehyde (increasing the percentage did not vary the outcome) but not significant. As seen below, the histograms are quite similar. Microscope images were similar to those shown above.

This demonstrates the validity of the technique as described, without needing further fixation. However, we understand that the optional use of a fixation step can be beneficial for clinical routine procedures or even when the lncRNAs are known to be in extremely low concentrations in the cell. This is why we added an optional fixation step to our RNA-protocol. We thank the reviewer for pointing this out as we believe this is of great interest in the clinic, as samples can be gathered routinely to then be analysed at the same time.

Page 8 lines 245 to 246

Please note: A fixation step can be performed prior to staining when required (see protocol below)

Page 8 lines 263 to 267

Optional fixation step when required: Cell pellets were resuspended in 2% Formaldehyde in 1xPBS for 10 minutes at room temperature. Continue by adding 2 volumes of 1xPBS to dilute the Formaldehyde and centrifuge at 260g for 5 minutes. Discard supernatant and wash with 1 ml of PBS and continue with adding the hybridisation buffer (see above).

2. RNase treatment is not enough to confirm specificity of the probes. They should show decrease of telomere signals upon knock-down of TERRA using conventional high-resolution microscopic images.

We knew RNase experiments were not enough and this is exactly the rationale behind the use of Gapmers and its comparison to the q-PCR technique as the gold standard for RNA expression quantification (and up to know the only method to quantify TERRA). However, we have performed the experiments again to demonstrate the validity of the technique after GapmeR

knock-down also using confocal microscopy. For this we run RNA-Flow in U2OS cells 24 hours after TERRA knock-down with the specific GapmeRs. As previously seen in cytometry, using high-resolution microscopy we detected the reduction in TERRA signal intensity. Images were acquired with a confocal microscope in 5 different fields using stacked images for scramble and anti-Terra transfected cells. Quantification (Supplementary figure B) was performed using the Z-stack projection of the whole nuclei (80 images) whilst the picture in supplementary figure A is of a selected image (in the middle, image 25) for both conditions.

We thank the reviewer as we believe that the manuscript is clear now. This is now Supplementary figure 8.

Supplementary figure 8: Confocal images of the transfected U2OS with either the scramble GapmeR or the anti-TERRA GapmeR after staining with the RNA-Flow protocol. A) Left panels are the DAPI staining and the right panels show the TERRA staining with Cy3-PNA probe anti-TERRA lncRNA in an specific image of the Z-stack lot obtained. Image is the 25th out of 80 for both samples, scramble (control) and anti-TERRA. B) Quantification of the confocal images of the transfected cells. Z-stacks of 80 images were captured in 0.47 µm sections across the entire nucleus with a Nikon A1R laser scanning confocal microscope (Nikon) using a 60X/1.4 Plan-Apochromat objective. Acquired Z-stacks were reconstructed (sum intensity projection) and analyzed using Fiji v1.53.

3. The manuscript does not follow standard scientific format, lacking clear introduction and result sections. I understand Comm Biol accept wide variety of format at a initial submission, but the current format is not acceptable, at least for me.

We have reformatted the article according to the journal's guidelines and the comments written by the reviewer. We do not pinpoint all the changes, as the whole manuscript was adapted to formatting guidelines. We chose the structure: 1) introduction, 2) results and discussion and 3) methods, as we believe it better fits the manuscript.

Reviewer #2 (Remarks to the Author):

González-Vasconcellos et al. described a new assay for quantifying intracellular levels of lncRNAs TERRA and TERC using flow cytometry and peptide nucleic acid probes. They provided solid demonstration of the assay's functionality. I only have two questions/suggestions for minor revision. Once these questions are addressed (if possible), the article will make an interesting read for the field of scientists working on lncRNA biology.

1. The authors used 0.5×10^6 cells for experiment. Could you comment on the effect of using higher or lower cell load?

We thank the reviewer for pointing this critical issue out, as sometimes gathering half a million cells is complicated specially in the clinic. For this we have performed the experiments on a range of cell numbers often used in flow cytometry experiments. We analysed in parallel and under the same probe concentration three different cell load:

We initially used 0.5×10^6 cells because it is a usual cell load in flow cytometry assays. However, we now tried: 0.25×10^6 ; 0.5×10^6 ; and 1×10^6 cells. The results are plotted in the new supplementary figure 4. We observed very little differences on the RNA-Flow outcome for the three different cell amounts.

We believe this strengthens the method greatly and we have now added the cellular range to the protocol and now reads within the methods section:

Page 7 lines 239 to 241

Please note that cell loads between $0,25 \times 10^6$ and 1×10^6 can be used without altering the experimental outcome (supplementary figure 5).

2. The assay's applicability to peripheral blood cells can be of clinical utility. Could you describe in details how to process the blood sample: What anticoagulatives are compatible with the assay? What's the maximal acceptable transit time from blood draw to assay initiation (i.e. can leftover blood from say routine CBC or biochemistry tests be used)? How you retrieved the cells from a tube of blood?

We thank the reviewer for such an interesting question when trying to apply this method to the clinic and for having pointed out that the PMBC isolation was not stated in the manuscript.

We have extracted blood preserved either in heparin or EDTA and both worked for PBMCs isolation and RNA-Flow. Ficoll gradient was performed within the 24 hours after blood extraction. Interphase with the PBMCs was collected. Cells were washed in PBS once and

either frozen in N₂ or used directly for the assay. The PBMCs cells used in this manuscript came for N₂ vials. Leftover blood from say routine CBC as you suggest would be an ok source to run this experiment. Regarding the maximal acceptable transit time from draw to assay: the assay can be performed right from extraction up to around 24 hours after. The issue must be avoiding RNA degradation to the best of the possibilities and this might differ depending on the target so this should be established for each lncRNA.

The following paragraphs were added to the text. We thank you for your assistance.

Page 7 Line 231 to 235

PBMC extraction: Whole blood preserved either in heparin or EDTA was used in this manuscript. Ficoll extraction was performed within the first 24 hours after extraction. Interphase with the PBMCs was collected. Cells were once washed in PBS and frozen in N₂. PBMCs cellular pellets were thawed and RNA-Flow was performed.. Please note: Freshly extracted cells also work for the assay.

Page 5 Line 170 to 175

These significant results demonstrate the technique's effectiveness and potential in clinical procedures. PBMCs isolation should be best performed within the first 24 hours after blood extraction and analyse via RNA-flow either right after extraction or fixate the samples (optional fixation step) and gather different samples to run the assay at a later timepoint.

Some typos were noted but did not affect the overall interpretation of the results:

1. Page 2, line 71: "RNase inhibitor" instead of "RNA inhibitor".

We specify that the product name is indeed RNase inhibitor, from Roche. It is used to avoid RNA degradation by RNases and it is key to perform RNA-Flow maximising results.

2. Page 16, line 573: "RNase concentrations" instead of "RNA concentrations"

We tested RNase concentrations (and it is therefore correct in the manuscript) to make sure we used enough of the enzyme to prove that the target of the assay is indeed RNA. Very low concentrations of RNase and the lack of RNase inhibitor was enough to abolish the signal.

3. Page 16, line 556: the sentence is grammatically confusing: "And making sure the reduction of the melting point due to the formamide in the hybridisation buffer does not infer cross reactions with the DNA."

We thank the reviewer for pointing this out and it has been deleted from the legend as it was not necessary and it is properly explained in the manuscript.

Reviewer #3 (Remarks to the Author):

The study by Gonzalez-Vasconcellos et al. presents a new RNA-flow method using PNA probes to detect and quantify TERRA and TERC RNA by flow cytometry. They optimized the

hybridization conditions, RNase inhibitors, and probe concentration. Three cell lines (U2OS, MDAMB231, and HepG2) were initially tested using this method and the results were consistent with the qRT-PCR data. They nicely showed that the reduction of TERRA RNA by knockdown can be confirmed using this method. TERRA expression during cell cycle progression was analyzed by RNA-flow and it displayed an increase of TERRA level in the mid-S phase. They demonstrated that this method can apply to human periphery blood mononuclear cells (PBMC). Similar experiments were also performed for TERC RNA quantification using a Tel-G PNA probe. This RNA-flow method could be very beneficial for clinical assay and basic research. However, a high probe concentration used in the hybridization and no pause points in the protocol can be a drawback. The major concerns below are needed to be addressed before publication.

Major concerns

1. Does this RNA flow method have to be done in freshly collected cells? Any steps can be paused before hybridization?

We thank the reviewer for this questions and we have amended the protocol adding an optional fixation step using PFA (paraformaldehyde at 2%) as you can see below in the fixation question.

If the researcher or clinician chooses to fixate, this can be stopped to be continued in the following days. However, unlike other RNA techniques, from collection to hybridization with this technique you need about 20 minutes of bench time which makes it faster than most protocols available without needing a fixation step although, it can be performed if necessary.

2. The optimal probe concentration is 300 nM, which is quite high for RNA flow. The authors collected cells and performed hybridization without fixation and permeabilization steps, thereby the probes may not efficiently enter the cell. Duckworth et al., (Nature protocol, 2019) have described a method to detect RNA and proteins using flow cytometry and the probe concentration is in the range from pM to nM. Porichis et al., (Nature Comm, 2014) also used a branched DNA method to amplify RNA signals to detect RNA by flow cytometry (now is available in ThermoFisher Scientific, called PrimeFlow RNA assay).

We thank the reviewer greatly for pointing out this mistake we made in the writing of the manuscript (It was properly written in the figures). The probe has been used in the nM concentration and we have corrected this throughout the manuscript and figures. Again let us thank you for finding this critical point in the manuscript.

Both methods include fixation and permeabilization steps. Do the authors try the fixation and permeabilization steps before hybridization? Can authors compare their method and others' RNA flow methods to show its advantages and disadvantages?

We checked the cells right after staining (following the RNA-Flow protocol without any extra fixation, just the cell suspension we used for cytometry) in the microscope and we show the picture below. The DAPI staining shows intact nuclear structure without DNA fragmentation or nuclei bursting and the TERRA signals are perfectly visible within the nuclei demonstrating correct cellular permeabilization and targeting of TERRA. In fact, these results are the same as those obtained using conventional RNA-FISH.

Figure 1: U2OS cell stained with the RNA-Flow protocol seen under the confocal microscope. The DAPI image showed intact nuclei. PNA staining of TERRA transcripts with Cy3 show a dotted pattern within the nuclei with different sizes within each nuclei. The image is a representative Z-stag (stag 25 in the middle of the nuclei).

However, to eliminate any shadow of doubt about the protocol, we performed the RNA-Flow experiment after fixation with 2% and 4% PFA for 10 minutes prior to the PNA staining. We found a slight improvement on the mean fluorescence after 2% Formaldehyde (increasing the percentage did not vary the outcome) but not significant. As seen below, the histograms are quite similar. Microscope images were similar to those shown above.

This demonstrates the validity of the technique as it is described without needing further fixation. However, we understand that the optional use of a fixation step can be beneficial for clinical routine procedures or even when the lncRNAs are known to be in extremely low concentrations in the cell. This is why we added an optional fixation step to our RNA-protocol. We thank the reviewer for pointing this out as we believe this is of great interest in the clinic, as samples can be gathered routinely to then be analysed at the same time.

Page 7 lines 245 to 246

Please note: A fixation step can be performed prior to staining when required (see protocol below)

Page 7 lines 263 to 267

Optional fixation step when required: Cell pellets were resuspended in 2% Formaldehyde in 1xPBS for 10 minutes at room temperature. Continue by adding 2 volumes of 1xPBS to dilute the Formaldehyde and centrifuge at 260g for 5 minutes. Discard supernatant and wash with 1 ml of PBS and continue with adding the hybridisation buffer (see above).

The main advantages of this technique are:

- 1) It is the first method to allow total TERRA expression levels
- 2) No need for an extra fixation step (although it can be performed).
- 3) Short bench time: The whole protocol from cellular pellet to staining of about 15 minutes
- 4) Increased specificity: PNA probes are known for their high specificity thanks to the combination of the nucleic acid with the peptides. This also allows co-staining of several lncRNAs at the same time using different fluorochromes.
- 5) Increased quantification capabilities: due to the usage of the cytometer.

3. PNA probe (TelG) for TERC can potentially target antisense TERRA transcripts. Why do the authors use specific PNA probes for TERC? The TelG probe only contains 11 nucleotides complementary to TERC. Is this specific enough for TERC RNA? Or this should be discussed.

TERRA molecules are transcribed from the subtelomeric regions towards chromosome ends by RNA Pol II using the telomeric C-rich strand as the template. Azzalin C.M., Lingner J. Telomere functions grounding on TERRA firma. *Trends Cell Biol.* 2015; 25:29–36. doi: 10.1016/j.tcb.2014.08.007. Together with this, it is known that TERRA binds TERC by using the sequence recognition site that we use in this manuscript. (Chu HP, Cifuentes-Rojas C, Kesner B, et al. TERRA RNA Antagonizes ATRX and Protects Telomeres. *Cell.* 2017;170(1):86-101.e16. doi:10.1016/j.cell.2017.06.017). Together with this, we have developed specific GAPmers designed specifically against the TERC sequence to demonstrate the binding of the probe.

Minor concerns

1. “RNase” is commonly used instead of RNase. Please replace it throughout the manuscript. Replace “DNase” with “DNase” throughout the manuscript.

We thank the reviewer for pointing this out. However, we have checked in the product data sheet, and well as in several writing sources and it is all written as we wrote it in the manuscript. RNase and DNase.

2. Please write in a symbol for the Celsius scale to replace “degrees” throughout the whole manuscript

This issue has been addressed throughout the manuscript. Thank you.

3. Which RNase was used in Supplementary 4, 5? Please specify it. RNase A or a cocktail of RNases?

This was the RNase from Roche Ref: 10109134001, which is a crude mixture of RNases. We have added this information to the manuscript within methods. We thank the reviewer for pointing out this lack of important information.

Page 8 lines 247 to 249

The RNase treatment was performed by adding into the labelling hybridisation buffer 50 µg crude mixture of RNases (DNase free) from bovine pancreas (Roche)

4. Line 280" there is a typo in 250.000 cells. It should be 250,000 cells.

Corrected thank you.

5. Line 113: Missed a comma in the sentence. To further document the specificity at targeting this specific lncRNA, the TERRA transcripts in U2OS cells were

Addressed thank you

6. Line 187: a typo in "fetal" bovine serum

Addressed thank you.

Reviewers' comments:

Reviewer #1 (Remarks to the Author):

Authors answered all the questions raised by this reviewer.

Reviewer #2 (Remarks to the Author):

My comments and questions were properly addressed. I do not have further questions. I believe the article will make an interesting paper to the journal's readership.

Reviewer #3 (Remarks to the Author):

In the revised manuscript, several issues have not been resolved. Although the RNA-flow method without fixation could save some bench time, the fixation may improve the nuclear structure and have better staining of RNA-flow. The comparison of fixation and non-fixation should be done and shown in the main figure.

See below

1. Please show the RNA-flow data using 2% formaldehyde fixation. The nuclear structures are distorted (irregular and unsmooth nuclear shapes in the DAPI image) when using the original protocol without fixation. Therefore, fixation seems to be a better way to preserve samples. Therefore, this is important data that should be included in the main figure. The TERRA intensity is slightly higher in fixed samples when compared to the original protocol without fixation.

2. PNA probe (TelG) can perfectly target antisense TERRA transcripts. The TelG probe only contains 11 nucleotides complementary to TERC. I still have doubts about the specificity of TelG probe for TERC detection.

The TERC knockdown experiment cannot rule out the probe specificity. This might be due to the low level of antisense TERRA transcripts. Although TelG may target TERC, it is also perfectly complementary to the antisense TERRA. They should use TERC specific probes with perfect matched sequences and check the colocalization with TelG PNA probes.

3. Line 81: TelC probe contains (TAACCC)_n repeats instead of (AATCCC)_n. Please write the sequence in 5' to 3' direction.

4. I have checked the RNase inhibitor product sheet from Roche and other companies. They stated, "RNase" rather than "RNase". Most papers nowadays have commonly used "RNase". Please revise them through the text.

<https://www.scientificlabs.co.uk/product/3335402001>

Line 86, 87, 98, 117,119,123 and so on.

5. Please write in a symbol for the Celsius scale to replace "degrees" throughout the whole manuscript.

Line 93, 94

6. Line 321: "300.000" U2OS. Is this a cell number?

7. Line 337: 2500.000 cells? Should be 250,000 cells. Line 277: 500,000 cells. Not 500.000 cells. There are more mistakes of "," and "." for the cell numbers. Please correct all those throughout the manuscript. Line 292,242,243

8. Line 255, 258,262,263: 0.1 % tween 20. Not 0,1 %

Reviewer #3 (Remarks to the Author):

In the revised manuscript, several issues have not been resolved.

Although the RNA-flow method without fixation could save some bench time, the fixation may improve the nuclear structure and have better staining of RNA-flow. The comparison of fixation and non-fixation should be done and shown in the main figure.

We truly thank again the reviewer for the very useful comments. Although tough to resolve, some of them were certainly key to shaping the final version of the manuscript. Please have a look at all the answers and modifications we have undertaken to give proper answers to your very useful comments and doubts. We hope this second review clarifies your doubts and satisfies your review process. We give an answer to each point below.

1. Please show the RNA-flow data using 2% formaldehyde fixation. The nuclear structures are distorted (irregular and unsmooth nuclear shapes in the DAPI image) when using the original protocol without fixation. Therefore, fixation seems to be a better way to preserve samples. Therefore, this is important data that should be included in the main figure. The TERRA intensity is slightly higher in fixed samples when compared to the original protocol without fixation.

We have performed this RNA-Flow protocol and specifically checked the cells under the microscope (cell suspension was placed in a chamber slide without any further processing) as asked by the reviewer.

We need to clarify that **the distortion seen in figure supplementary 8 from the previous rebuttal has to do with GapmeR transfection** (scramble or anti-TERRA) rather than with the RNA-Flow itself as the new experiments with non-transfected cells showed proper nuclear integrity as you can see below (Supplementary figure 6). Seeing the reviewer's fair doubt about the cellular integrity and the staining itself **we decided to show the confocal images in the main figures for both probes**. The figures are now in main figures 1C and 2B.

However, **specifically regarding the fixation**: when assaying under the microscope the TERRA signal without fixation and with 2% formaldehyde we realised that the apparent “better TERRA signal” seen in the cytometer after fixation we showed in the previous rebuttal is due to a slightly increased unspecific nuclear background. We checked with our microscopy experts and they corroborated that the fixed cells show a bit more unspecific background inside the nuclei and less spotted pattern (please see below in the actual supplementary figure 6).

Supplementary figure 6: Pre-fixation study for TERRA staining A) Confocal images of U2OS after TERRA staining using the RNA-Flow staining protocol. Left panels are the DAPI staining showing nuclear integrity and the right panels show the TERRA staining with Cy3-PNA probe anti-TERRA lncRNA. The upper panels were stained with no previous pre-fixation and lower panels were pre-fixed with 2% formaldehyde. Z-stacks of 80 images were captured in 0.47 μm sections across the entire nucleus with a Nikon A1R laser scanning confocal microscope (Nikon) using a 60X/1.4 Plan-Apochromat objective. Please note, the white square on the Cy3.TERRA upper panel shows the region of interest (ROI) used in figure 1C. The scale bar is 50 μm . B) RNA-Flow of the same U2OS cells. Cytometry data of the RNA-Flow TERRA staining. Black peak shows Cy3-Terra staining with no pre-fixation and the red peak shows the fluorescence intensity of the staining after a pre-fixation with 2% formaldehyde.

Therefore, we still believe that not only the unfixed protocol is faster but rather better at showing best the specific signal. We will add this picture to the manuscript in the supplementary material (supplementary figure 6) and keep the fixation protocol as before “optional and depending on the target and probe” if this is ok with the reviewer. Definitely, the protocol can be performed with and without fixation and it should be chosen upon specific needs. Both options were made available in the manuscript for the reader.

Furthermore, when it comes to the new specific TERC probe, no significant difference in the staining or intensity was found between using or not pre-fixation. This is stated in the manuscript in a new supplementary figure (supplementary figure 7) adding this data. In the case of this probe, the background did not increase, but the staining was comparable after assaying 10 different

Supplementary figure 7: TERC probe details A) Fixation study. Confocal images of MDA-MB231 after TERC staining using the RNA-Flow staining protocol. Left panels are the DAPI staining showing nuclear integrity and the right panels show the TERC staining with a specific Cy3-PNA probe. The upper panels were stained with no previous pre-fixation and lower panels were pre-fixed with 2% formaldehyde. Z-stacks of 80 images were captured in 0.47 μm sections across the entire nucleus with a Nikon A1R laser scanning confocal microscope (Nikon) using a 60X/1.4 Plan-Apochromat objective. Scale bar is 25 μm .

We truly believe upon this new evidence that the RNA-Flow does preserve nuclear/cellular structures without needing fixation and this is an important feature of this new protocol. However, it can be interesting for specific assays or probes and we describe this step as optional in the protocol again if the reviewer agrees with us.

2. PNA probe (TelG) can perfectly target antisense TERRA transcripts. The TelG probe only contains 11 nucleotides complementary to TERC. I still have doubts about the specificity of TelG probe for TERC detection. The TERC knockdown experiment cannot rule out the probe specificity. This might be due to the low level of antisense TERRA transcripts. Although TelG may target TERC, it is also perfectly complementary to the antisense TERRA. They should use TERC specific probes with perfect matched sequences and check the colocalization with TelG PNA probes.

We firstly should state again that TERRA molecules are transcribed from the subtelomeric regions towards chromosome ends by RNA Pol II exclusively using the telomeric C-rich strand as the template as this publication shows with a northern blot analysis: Azzalin C.M., Lingner J. where they describe:

“This approach revealed the existence of TERRA ranging in size from ~100 bases up to at least 9 kilobases (Fig. 1, A and B). The TERRA signal was abolished upon ribonuclease (RNase) treatment, confirming that it originated from RNA (Fig. 1A). TERRA molecules are found exclusively in nuclear fractions and contain UUAGGG repeats. We only detected a faint signal for the complementary CCCUAA repeats (Fig. 1A), suggesting that CCCUAA-containing RNA molecules might exist only at very low levels” Azzalin, C. M., Reichenbach, P., Khoriantuli, L., Giulotto, E., and Lingner, J. (2007). Telomeric repeat containing RNA and RNA surveillance factors at mammalian chromosome ends. *Science* 318, 798–801. doi: 10.1126/science.1147182

A similar statement is made here: Telomere functions grounding on TERRA firma. *Trends Cell Biol.* 2015; 25:29–36. doi: 10.1016/j.tcb.2014.08.007.

And this other paper demonstrating that TERRA binds TERC by using the sequence recognition site that we use in this manuscript with the TEL-G probe. (Chu HP, Cifuentes-Rojas C, Kesner B, et al. TERRA RNA Antagonizes ATRX and Protects Telomeres. *Cell.* 2017;170(1):86-101.e16. doi: 10.1016/j.cell.2017.06.017). So these papers pointed to the potential targeting of TERC by the telomeric probe (TelG). Adding to this, we designed specific Gappers against TERC, showing that the probe does indeed target the lncRNA.

However, we understand that the reviewer could be concerned by the potential unspecific signal raised by using this probe under RNA targeting conditions in this specific detection protocol.

This is why we developed, in partnership with PANAGENE a new PNA probe specific for human TERC: Cy3-OO-AGCAGCTGACATTTTTTGTGG.

Before we continue, I would like to point out that this does not affect the validity of the RNA-Flow method but rather **improves specifically the detection of TERC.**

Firstly, we performed a colocalization experiment with the TELG and the new TERC probe with the following outcome: If we specifically looked for nuclear colocalization, the colocalization index was ~80%, however, using the whole-cell, the colocalization coefficient decreases to around 40%. With this data we realised that the TELG probe potentially marks something other than TERC in the cytoplasm and that **the reviewer is right when assuming that the Tel G probe, although detecting TERC, does not exclusively target this long non-coding RNA and therefore the probe is**

not specific enough. TELG has a stronger signal in the cytoplasm when compared to the new TERC probe as you can see below.

Colocalization experiment figure: Failed colocalization experiment between TELG and the specific probe for TERC. Within the nuclei, colocalization was quantified at around 80% but whole-cell colocalization was never greater than 40%. The TEL-G probe has been discarded and all TERC experiments were performed with the specific probe named Cy3-TERC.

We thank the reviewer for detecting this issue and for suggesting checking our cells using microscopy as we truly believe the manuscript has profited from such an idea.

Having developed the new probe, we believe it improves the specificity of TERC detection. In fact, this increases the importance of the paper as it is the first PNA probe specifically designed and made available for TERC (same as with the GapmeR).

Manuscript changes:

- 1) We run all the TERC probe related experiments shown in the manuscript figure 2 again with the specific probe and changed the figure and manuscript accordingly. We demonstrate that the RNA-Flow works with this probe, also show the confocal microscopy, GapmeR knockdown and the study on PBMCs.

Changes on figure 2: figure 2A (TelG binding scheme has been removed), 2B has been performed with the new probe, 2B (microscopy) has been added and, 2C (cellular quantification) has been reproduced with the new probe, 2D (PBMCs) reanalysed with the new probe and the GapmeR knockdown has also been replaced by the one with the new probe. You can see below the new figure 2.

- 2) All changes made in the manuscript are tracked in red in the main document including the fixation explanation, the new probe and the microscopy work.
- 3) Figure 1 now shows the microscopy images of TERRA staining using RNA-Flow staining protocol.
- 4) Supplementary figures 6 and 7 were added.

3. Line 81: TelC probe contains (TAACCC)n repeats instead of (AATCCC)n. Please write the sequence in 5' to 3' direction.

Corrected. Thank you.

4. I have checked the RNase inhibitor product sheet from Roche and other companies. They stated, "RNase" rather than "RNAse". Most papers nowadays have commonly used "RNase". Please revise them through the text.

<https://www.scientificlabs.co.uk/product/3335402001>

Line 86, 87, 98, 117,119,123 and so on.

Corrected, thank you.

5. Please write in a symbol for the Celsius scale to replace “degrees” throughout the whole manuscript.

Line 93, 94

Corrected, thank you.

6. Line 321: “300.000” U2OS. Is this a cell number?

Corrected, thank you.

7. Line 337: 2500.000 cells? Should be 250,000 cells. Line 277: 500,000 cells. Not 500.000 cells. There are more mistakes of “,” and “.” for the cell numbers. Please correct all those throughout the manuscript. Line 292,242,243

Sorry about these mistakes hindered by having the computer settings in Spanish...corrected, thank you.

8. Line 255, 258,262,263: 0.1 % tween 20. Not 0,1 %

Corrected, thank you.

The new version of the paper is now ready to be revised again by the reviewer and we hope it satisfies his standards and those of the editor too.

Thank you again and we look forward to hearing from you both regarding this rebuttal.

REVIEWERS' COMMENTS:

Reviewer #3 (Remarks to the Author):

The revised manuscripts answered the major concerns. There are some minor issues that the authors did not correct them such as "RNase" labels in several figures (Figure 1D, Figure 2C, Supplementary Figure 1, 4) and cell numbers in the text (Line 282, 305, 316).

Please use "RNase" instead of "RNAse"

Line 282: "0.5x10⁶" cells rather "0,5x10⁶"

Same as Line 305 and 316.